# Effects of Fine-Mesh Exclusion Netting on Pests of Blackberry

**DOI:** 10.3390/insects10080249

**Published:** 2019-08-14

**Authors:** Ryan Kuesel, Delia Scott Hicks, Kendall Archer, Amber Sciligo, Ricardo Bessin, David Gonthier

**Affiliations:** 1Department of Entomology, University of Kentucky, Agricultural Science Center, S-225, Lexington, KY 40508, USA; 2The Organic Center, The Hall of the States, 444 N. Capitol St. NW Suite 445A, Washington, DC 20001, USA

**Keywords:** organic management, spotted-winged Drosophila, *Drosophila suzukii*, green June beetle, *Cotinis nitida*, Japanese beetle, *Popillia japonica*, row covers, fine-mesh exclusion netting

## Abstract

Fine-mesh exclusion netting is a potential alternative to organic and conventional insecticide application to control numerous pests of fruit crops. We tested whether fine-mesh exclusion netting would reduce pest abundance and increase marketable yield compared to organic spinosad insecticide sprays in an organically managed blackberry field. At the completion of flowering, we covered blackberry rows with fine-mesh exclusion netting (ProtekNet) and treated alternating rows with an organic spinosad insecticide (Entrust™). Fine-mesh exclusion reduced green June beetle (*Cotinus nitida* Linnaeus) and bird presence and marginally reduced Japanese beetle (*Popillia japonica* Newman) presence on blackberry canes compared to organic spinosad insecticide treatment. Exclusion netting reduced the capture of spotted-wing Drosophila (*Drosophila suzukii* Matsumara; “SWD”) in baited traps in the fourth week of exclusion and reduced the overall number of SWD adults emerging from harvested blackberry fruits. Marketable yield in the fine-mesh exclusion treatments was two times higher than the organic spinosad insecticide treatment. These results suggest that fine-mesh exclusion netting is a functional pest control alternative to insecticide treatment for organic blackberry production.

## 1. Introduction

Caneberry growers struggle to meet the demands for fresh, local, and organic fruit due to strong pest pressure, which is a challenge to manage without the use of insecticides [1,2]. Conventional insecticides in the pyrethroid, organophosphate, and spinosyn classes are efficient in controlling most insect pests of fruit crops, however, many endanger beneficial organisms like pollinators and natural enemies of pests [3]. Even organic registered insecticides are broad spectrum and can have lethal and sub-lethal consequences for beneficial organisms [4,5]. Because of this, organic certification suggests that farmers use chemical pest control (biological or botanically-based) only after systems-based non-chemical strategies have been employed and have failed to achieve sufficient pest control. Therefore, organic farmers need effective alternatives to organic insecticides [6]. Indeed, given that a number of fruit crops, including caneberries, depend on pollination and natural enemies to improve yield or fruit quality, growers and researchers continue to develop and improve pest-control strategies that are effective alternatives to insecticidal sprays [7].

Cultural practices, physical barriers (especially mesh netting exclusions), introduction of biological control agents, and the use of plant-derived deterrents are current alternatives to insecticides that hold promise for pest control. For instance, the planting of early ripening crop varieties can help growers avoid infestations of pests if the fruiting period does not overlap with the period of pest populations [8]. However, this may limit grower access to certain late season markets that may earn a premium for berries. Alternatively, sanitary measures can be undertaken and include the removal of nearby, wild, fruiting plants and groundcover vegetation as well as the collection of dropped or leftover fruit in order to reduce food and host plant resources available to pests [9].

Fine-mesh netting exclusions act as physical barriers to limit insect pests from consuming above-ground vegetative material and fruits. In vegetable crops, ‘floating row covers’ or ‘meso-tunnels’ have been used to cover individual or multiple rows with fine-mesh materials that limit insect access to plants [10,11,12,13,14,15]. Fine-mesh netting exclusions have also been applied in grape [16], blueberry [17,18,19], and raspberry [20,21]. Indeed, the adoption of fine-mesh nets may be particularly feasible for producers of grapes, caneberries, and blueberries who already employ course-mesh netting to prevent frugivorous birds from consuming their crop [22]. For these growers, fine-mesh nets could be easily substituted for bird netting without any additional cost for netting infrastructure. Fine-mesh netting would still successfully exclude fruit-eating birds and could provide the added benefit of suppressing insect pests. Additionally, in cropping systems where low thresholds of pest activity are required due to the high risk of disease transmission or food safety concerns, netting may be especially promising. The production of bacterial wilt-prone curcubitaceous crops, such as muskmelon and acorn squash, are examples of some of the current inquiries into this topic [10,11,12]. When examined in the field, some variable benefits for crop yield and bacterial wilt incidence were reported when spun-bonded or fine-mesh exclusion nets were employed. In fruit and berry production in the southeastern region of the USA, however, there are still few field trials evaluating the efficacy of fine-mesh exclusion netting systems.

There are a number of important insect pests of caneberries common to the southeastern United States. Of particular importance is the invasive spotted wing Drosophila (*Drosophila suzukii* Matsumara; SWD), that is endemic to eastern Asia. During SWD’s initial infestation in 2008, USA west coast growers saw a combined loss of 16.4% of all blueberry, raspberry, blackberry, strawberry, and cherry (421.5 million USD) [23]. More recently, a survey of 82 raspberry producers in Minnesota estimated that both organic and conventional producers lose about 20% of their raspberry yields to SWD infestation [24]. They estimated that financial losses from SWD totaled about 11.8 million USD in raspberry production in that state. This statistic is interesting to us because Minnesota growers rely heavily on local, direct-to-market sales to make raspberry production profitable. This is quite similar to the production of blackberries in Kentucky. Crop losses of 20% in our state have the potential to force growers away from the production of caneberries towards other less vulnerable small fruits.

Female SWD have a unique ability to damage fruits using their highly sclerotized ovipositor that bears enlarged, tooth-like bristles [25]. Gravid female SWD show a distinct preference for oviposition within partially ripened fruits [9,26]. Other vinegar flies (*Drosophila* spp.) also infest damaged caneberries and can cause secondary damage. In addition to Drosophilid flies, the invasive Japanese beetle (*Popillia japonica* Newman) and the green June beetle (*Cotinis nitida* L.) are devastating pests of caneberries during mid and late summer. These beetles tend to swarm on fruiting plants, attracted by feeding-induced plant volatiles, and chew directly into fruits [27,28]. Finally, a number of bird species consume ripe fruits of caneberries by plucking individual drupelets from fruit clusters during harvest months [22].

In this study, we evaluated whether fine-mesh exclusion netting would limit damage from insect and bird pests and increase marketable yield in blackberry production in Kentucky. We compared fine-mesh exclusion netting systems to an organic standard insecticidal management system at field scale to provide a comparison of the two crop protection strategies. We predicted that fine-mesh exclusion netting would reduce levels of insect pests and increase marketable yield compared to the organic standard insecticidal system.

## 2. Methods

This study was conducted at the University of Kentucky horticulture research farm in Lexington, Kentucky in the summer of 2018 (plant hardiness zone 6b). This approximately 50-hectare research farm is surrounded by a landscape of urban and rural development. The farm is split into organic and conventional halves where a wide variety of perennial and annual crops are grown. Our research focused on a blackberry plot adjacent to conventional plots of apple orchard and wine-grape vineyards. The plot consisted of six rows, each containing three plants each of mature Triple Crown, Hull, and Chester varieties for a total of nine plants per row. The blackberries were planted with a 1.8 m tall trellis in rows with 1 m spacing within rows and 3.5 m wide alleys between rows. Each row had a length of 25 m. Varieties were planted within row in a randomized complete block design. In June of 2018, each row was hand-weeded, blackberry canes were trimmed back, and were then maintained on a trellis system. We chose to perform this experiment on a small-scale plot in order to imitate the small-scale production of the fruit found in Kentucky. In this state, most blackberry production consists of smallholder farms that grow blackberries in limited quantities for self-harvest (e.g., pick-your-own), local markets, or for use as ingredients in value-added products.

### Blackberry Fine-Mesh Netting Exclusion

In late June, we began an experiment with the goal of comparing the effectiveness of a fine-mesh exclusion netting system (ProtekNet) and an organic insecticide spray regime (Entrust™—Dow AgroSciences LLC, Indianapolis, IN, USA) for the control of multiple pests of blackberry. This organic spinosad is the most, and arguably the only, efficacious pesticide available to organic blackberry producers in the United States for the control of *Drosophila suzukii* [29]. We began this experiment after petal fall, when roughly 95% of blackberries had set fruit. This enabled us to exclude fruit-damaging pests without inhibiting pollination. Some fruits on treated blackberry canes had just begun to color, but none had yet begun to turn black, a stage in which they would be vulnerable to attack by insect pests. We netted complete rows to test the effectiveness of this pest control strategy on a scale representative of implementation by small-scale commercial growers. For this reason, and due to the size constraints of the available blackberry plot, we did not subset rows and did not include a no-management control. On alternating rows, we covered all nine blackberry plants in individual rows with a knitted polyethylene mesh ProtekNet (0.85 mm by 1.4 mm mesh size) 6.3 m wide, acquired from Dubois Agrinovation (Saint-Rémi, Quebec, QC, Canada). In each of the three netted rows, netting was held above the blackberry canopy by the existing t-frame trellis system and weighed down at ground level with cement pavers to protect against wind and to prevent intrusion from birds and insects (Figure 1). It should be known that the integrity of the exclusionary netting in two rows was compromised for a period of time shorter than 12 h due to a severe weather event which partially removed the netting on two rows. Uncovered rows acted as a representation of the standard organic protection regime. Each row was treated with spinosad applied three times in July using a backpack mist sprayer (Stihl SR450 Backpack Mist Sprayer, STIHL Inc., Virginia Beach, Virginia, VA, USA). We aimed to perform each insecticide application in roughly seven-day intervals. However, we allowed flexibility in this schedule and only performed pesticide sprays under low wind-speed-conditions to minimize drift of Entrust™ spray.

To determine the effect of the management systems on SWD, we placed baited traps (40 mL apple cider vinegar and 10 mL pure laboratory grade ethanol) in the center of each row directly within the blackberry canopy and underneath row covers. Baited traps were constructed from a lidded 473-mL red plastic drinking cup (Dart Container Corporation P16R, Mason, MI, USA). Two rows of twelve perforated holes encircled each cup at five and six centimeters from the base to act as semi-selective entrance points for arthropods. As an additional olfactory attractant, one pair of Pherocon SWD Peel-Pak, broad spectrum *Drosophila* lures was hung inside each trap (Product 5001-1P—Trécé Incorporated, Adair, OK, USA). Traps were attached to trellis posts at a height of 1.5 m and secured from wind-disturbance using zip-ties. Trap contents were collected weekly, then immediately refilled with bait solution. Broad spectrum lures were replaced once after 14 days, and trap data was collected for a total of four weeks across the month of July. The number of SWD, as well as the number of other vinegar flies per trap, were quantified under a stereoscope.

To compare the effects of the two management systems on SWD and all other vinegar flies collected in baited traps, we conducted linear mixed models (LMM) with management treatment (fine-mesh exclusion versus spinosad insecticide) and sampling week as fixed effects with the function ‘lmer’ in the R-package lme4 (Program R 3.5.1). Each individual trap was treated as a random effect to account for multiple repeated measurements across the weeks. Each proximal pair of rows was treated as a random effect to block row pairs together and reduce potential microclimatic biases. Vinegar fly data was then square root-transformed to achieve normality. We also uncovered a significant interaction between management treatment and sampling week by LMM analysis, so we conducted a paired t-test to compare the differences in SWD and vinegar fly presence within each of the four weeks.

To determine SWD infestation of berries during peak harvest, four weeks after initial netting, we collected samples of 20 overripe (non-marketable) berries per variety per row (three samples per row; nine per treatment). These berries were placed in an incubation chamber (Percival I-66VL two-door incubator, Percival Scientific Inc., Perry, IA, USA) inside of rearing containers (16 oz. Dart Solo MicroGourmet 16NW-0007, Mason, MI, USA) with mesh tops. The bottom of the rearing container was covered with sand and a 2.5 cm square sticky trap was suspended from the top of the container to immobilize emergent flies. After 20 days, we counted the number of SWD adults that emerged from the fruits under a stereoscope. To compare the effects of each management system, we conducted an LMM on log-transformed SWD emergence with management treatment as a fixed effect within the model. Each proximal pair of rows was treated as a random effect to block row pairs together. Each blackberry variety was treated as a random effect to account for any differences between varieties.

To document the abundance of Japanese beetles and green June beetles on blackberry canes, we used visual surveys. An observer, starting from one end of each experimental row, moved slowly under the exclusion netting or adjacent control plots, choosing the largest fruit-bearing cane per plant on six total plants per row. This was done in a way to minimize the disturbance of the beetles and to avoid double-counting individual beetles. Each selected fruit-bearing cane was scanned from top to bottom for the number of each beetle species present on fruits, stems, or leaves. This survey was conducted for each row on two occasions on 17 July and again on 25 July. To compare the effects of each management system, we log-transformed beetle abundance and conducted LMM with management treatment and sampling week as fixed effects. Each row was treated as a random effect to account for multiple repeated measurements across weeks. Each proximal pair of rows were treated as a random effect to block row pairs together.

At the completion of the netting experiment, once netting materials were removed, we walked each row and counted the number of bird fecal droppings on all leaves, fruits, and stems to estimate bird activity in rows with fine-mesh exclusions and organic insecticide treatments. To compare the effects of each management system on bird intrusion, we conducted a paired t-test with each proximal pair of rows as the grouping factor and compared the difference in bird droppings between the two management systems.

Following the ripening of fruits, we harvested blackberries each week for six weeks. We denoted unmarketable berries as those with damage or deformation and marketable berries as undamaged, ripe fruits fit for direct-to-market sale. We pooled yield measurements across weeks and across varieties within rows, then compared square root-transformed marketable, unmarketable, and total yields with LMM. Management treatment was our fixed effect within the model. Each proximal pair of rows was treated as a random effect to block row pairs together.

To assess the quality of marketable yield, we compared the sugar content of berries with a PAL-Easy ACID4 pocket sugar and acid meter two times immediately after harvest in early and late July (Atago Co., Ltd., Tokyo, Japan). We compared the sugar content of marketable berries with LMM with management treatment as a fixed effect and row pair, sampling week, and blackberry variety as random effects within the model. For all LMM analyses performed in this study, we confirmed the normality of each set of data with a Shapiro–Wilk normality test of model residuals.

## 3. Results

### 3.1. Spotted Wing Drosophila and Other Vinegar Flies in Baited Traps

Data analysis using linear mixed models revealed a significant interaction between sampling week and management treatment in baited traps when calculated across all four trapping weeks (Table 1A, Figure 2A). This interaction suggests that our two treatments of fine-mesh exclusion versus organic insecticide sprays affected the presence of spotted wing Drosophila inside each row quite variably across each of the four weeks. Very low numbers of SWD were trapped during the first three weeks, as only four total SWD individuals were trapped during this time. There was no significant difference between exclusion treatments in week one (t = 1.00, *p*-value = 0.4226), week two (t = 1.00, *p*-value = 0.4226), nor week three (t = 2.00, *p*-value = 0.1835). However, in week four, there were significantly fewer SWD in fine-mesh netting exclusion rows compared to organic insecticide treated rows (t = 4.44, *p*-value = 0.0472) (Figure 3B).

Other non-SWD vinegar flies were present in high numbers across all four weeks (Figure 2B). Nonetheless, a significant interaction between sampling week and fly count was again observed (Table 1B). Individual week analysis revealed significantly lower vinegar fly captures in exclusion rows compared to insecticide treated rows for week one (t = 5.23, *p*-value = 0.0347) and week two (t = 5.50, *p*-value = 0.0315), but non-significant effects in week three (t = 2.63, *p*-value = 0.1195) and week four (t = 1.65, *p*-value = 0.2414).

### 3.2. SWD Emergence from Overripe Berries

Adult SWD emergence from over-ripened blackberry fruits was significantly lower in exclusion treatments than in insecticide treatments (Table 2, Figure 3C). On average, blackberries in organic insecticide treatments contained nearly 31 times the number of SWD flies per fruit than fruits under fine-mesh netting treatments.

### 3.3. Japanese Beetle, Green June Beetle, and Bird Presence

Green June beetle and bird intrusion were lower in fine-mesh netting exclusion treatments compared to the organic insecticide treatments (Table 2, Figure 3A). The number of Japanese beetles present tended to be lower in exclusion treatments, but did not reach statistical significance at a 5% confidence interval (Table 2, Figure 3A).

### 3.4. Yields

The yield of marketable blackberries was on average 2.04 times higher for fine-mesh netting exclusion treatment (210 g/row/week) relative to organic insecticide treatment rows (103 g/row/week) (Table 2; Figure 4A). The total yield of blackberries was on average 1.79 times higher for fine-mesh netting exclusion treatment (281 g/row/week) relative to organic insecticide treatment rows (157 g/row/week). Unmarketable yields did not differ between treatments (Table 2, Figure 4A).

### 3.5. Blackberry Quality–Sugar Quantity

Sugar content of blackberry fruits tended to be higher in fine-mesh netting exclusion treatments, but this effect did not reach statistical significance at a 5% confidence interval. Average sugar content of blackberries in exclusion treatment was 8.39 °Bx and the average sugar content in organic insecticide treatments was 7.28 °Bx (Table 2, Figure 4B).

## 4. Discussion

From this study, we showed that spotted wing Drosophila emergence from harvested blackberries was significantly reduced by exclusionary netting. This fine-mesh exclusionary treatment was therefore more successful at reducing SWD infestation of fruits than an organic spray regime of the spinosad Entrust™. Still, SWD did succeed in accessing netted fruits in two of three rows and was shown to have infested fruits in one of these rows. We suspect that this limited infestation of netted rows occurred due to the temporary removal of the exclusion cover caused by an extreme weather event. Therefore, we now strongly recommend that small fruit producers who choose to implement fine mesh exclusions for control of SWD should deploy baited traps in order to detect SWD intrusion. If SWD do intrude, pesticidal sprays beneath exclusion netting could allow the limitation of infestation prior to breeding and population expansion. The delayed appearance of damage in blackberries caused by infestation of SWD presents a new threat to caneberry production. For this reason, new management strategies need to be developed that are easy to implement and rapidly deployable in response to the seasonal arrival of SWD. The production of fruit crops in small scales requires flexible control measures that are cost-effective and functional.

Ultimately, SWD population sizes at our experimental location were very low, at least until the end of our third trap collection, and nearly 75% of the year’s blackberry harvest was collected before then. This indicates that treatment for SWD may only have been necessary for the tail end of the harvest at this location. This near miss for SWD pressure opens the door for future varietal trials to examine fruiting phenology of additional blackberry varieties. If varieties exist that complete full fruiting one or two weeks sooner, it is possible that damage from SWD can be avoided completely in climatically favorable years. From additional sampling of SWD populations in Kentucky, early blueberry production in some years may already escape SWD pressure by concluding before SWD populations reach damaging levels. However, this capability has been shown to be true of only some varieties of blueberries in geographically disparate regions of the United States [8]. This phenological occurrence is most likely strongly dictated by the variable climate of each year as well, and some years may yield large populations of SWD during blueberry as well as blackberry fruiting.

While fine-mesh exclusion netting was effective for the prevention of SWD infestation, it also provided additional protection against beetle and bird pests of blackberry. This management strategy, therefore, provides non-chemical, broad spectrum pest control to berry growers. The feeding of Japanese beetles on blackberry canes was nearly entirely prevented by exclusion netting, as only one beetle was noted inside the netting treatment in each scouting period. Due to small sample size, this difference did not achieve statistical significance at a 5% confidence interval (Table 2). Green June beetle and bird presence within netted rows was never noted during each scouting period nor on any subsequent occasion during fruit harvest. The presence of all three pests was thus significantly lower within exclusionary rows than spinosad treated rows. Damage from all three of these pests is readily apparent on blackberry fruits during harvest, as beetles chew directly through fruit clusters, and birds tend to pick off individual drupelets. Both damage types result in completely consumed berries or damaged, rotting berry clusters that would not be harvested as marketable yields. As discussed in Ebbenga et al. (2019), this reduction in damage from non-SWD pests likely also prevents SWD adults from finding this required food source which could serve to slow SWD propagation when they do slip inside the exclusionary treatments [30]. Finally, marketable and total yields were significantly higher in exclusion rows, indicating the effectiveness of exclusion netting as protection against multiple types of pests.

We believe our work is the first within the literature to report on the use of fine-mesh exclusion netting for the protection of blackberries. Several studies have analyzed this method for control of SWD in other fruit crops, however, our study is among the first to show a significant increase in berry crop yield using exclusionary netting for pest control. McDermott and Nickerson (2014) showed a slight, non-significant yield increase over untreated controls on their New York orchard when fine-mesh netting was applied to entire rows of blueberry in 2013 [19]. Similarly, Cormier and Firlej (2015) showed this netting increased the overall size of blueberries but did not affect yield weights compared to spinosad insecticides in 2012 in Quebec, Canada [17]. Ebbenga et al. (2019) tested the production of Minnesota wine grapes under exclusion netting against the conventional pyrethroid, zeta-cypermethrin [30]. They also artificially inoculated half of their netted plots with adult SWD. The efficacy of their treatments varied across the two years of study and between sampling weeks, but taken in whole, netted grapes both with and without artificial SWD release showed 95% lower grape infestation than the untreated control. Due to the trend of positive evidence for the effectiveness of this pest control method, additional studies across a variety of crops and hardiness zones should be conducted to provide the scientific and farming communities with a better understanding of the possible function of fine-mesh exclusion netting in berry production.

Research into the efficacy of closing high tunnels with fine-mesh netting has also become popular for the production of red raspberries. Rogers et al. (2016) looked at this control option in Minnesota [21]. They found that a higher percentage of raspberries were harvested as marketable fruits and fewer fruits were infested with SWD when grown under fine-mesh-sealed high tunnels or under high tunnels constructed entirely of fine-mesh netting, but that cumulative fruit yield did not differ by treatment. This was in comparison to berries grown in the open air with and without insecticidal control. Additionally, Leach et al. (2016) reported that Michigan raspberries, grown underneath mesh-sealed high tunnels, showed SWD infestation 10 days later than open treatments [20]. However, both netting and insecticidal sprays beneath the netting treatment was needed to better control SWD infestation of fruits than sprays alone. Further experimentation using this method is needed in order to continue to resolve the function of fine-mesh netting for production of berries beneath high-tunnels.

Exclusionary netting is a leading option for non-insecticidal control of pests in small fruit crops. Its use in commercial agriculture will be determined by its cost effectiveness, which will require further comparisons of input costs, yield benefits, and longevity of netting materials. Labor requirements for its installation may be costly at the start and end of production season as the netting is put up before fruit coloring and as it is removed at the end of fruit harvest. The durability of nets for on-farm use must also be analyzed in detail and may require individual farms to trial netting for their own independent scenarios. Statements from the netting manufacturer estimate the netting will last for five years. Studies published in the literature, however, place the longevity of similar fine-mesh nets from seven to ten years [20,31]. With careful upkeep, and short-term usage for only the five to eight-week period of fruiting, netting may indeed last for seven to ten growing seasons and could be a cost-effective investment for growers. Challenges to the adoption of exclusion netting may come from the steep learning curve of each grower’s first setup, the need to frequently remove or enter into exclusion areas for fruit harvest, and the possibility for tears to occur in the netting. If bird, beetle, or SWD pests achieve access to fruits through any of these windows of opportunity, the challenges of infestation or damage of fruit can rapidly manifest. Monitoring for these potential issues will remain a critical aspect of pest control using fine-mesh exclusion netting. Our finding that blackberry quality through sugar content is not significantly changed by use of exclusion netting is promising but should be studied in depth over multiple years and across a range of climates.

## 5. Conclusions

In conclusion, we have shown that fine-mesh exclusion netting functionally reduced the damage of some important pests of blackberry by restricting their access to fruits and foliage. This suggests that netting can be a viable alternative to organic insecticide treatment within the southern United States. Investigation into the overall profitability of this management system was beyond the scope of this study. Additional in-depth economic and agronomic analyses will be required to determine whether it is more profitable to switch from current organic pest control practices to exclusionary netting in blackberry production. Future studies should look to compare the costs of both the material inputs and labor required by both management schemes. Future studies could also investigate the use of fine-mesh netting exclusion for late-blooming varieties of different berry species, where netting enclosures may need to be artificially inoculated with pollinator species. Additional repetitions of this work should look to place multiple baited SWD traps within each treatment at different heights from the soil in order to look at the spatial tendencies of adult SWD within netted and unnetted blackberries. Nonetheless, this study suggests that for organic growers, fine-mesh exclusion netting is a viable alternative to insecticide treatment.

## Figures and Tables

**Figure 1 insects-10-00249-f001:**
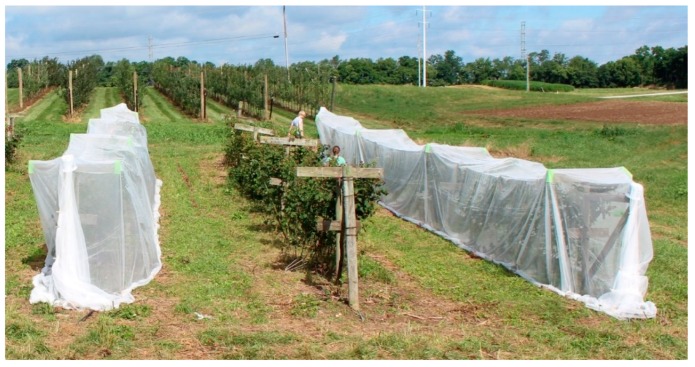
Photograph of experimental design. Pictured is two of three rows treated with fine-mesh exclusion netting. Also pictured is one of three rows treated with the organic spinosad Entrust™.

**Figure 2 insects-10-00249-f002:**
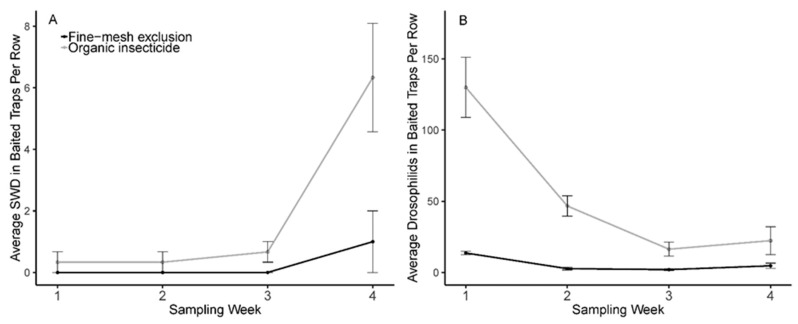
Baited trap captures of *Drosophila suzukii* and Drosophilids within blackberry rows. (**A**) Average number of SWD captured per row across four sampling weeks under fine-mesh exclusion and organic spinosad spray management; (**B**) Average number of non-SWD Drosophilids captured per row across four sampling weeks under fine-mesh exclusion and organic spinosad spray management.

**Figure 3 insects-10-00249-f003:**
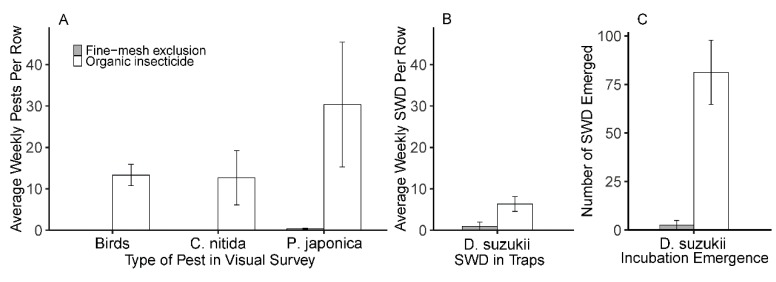
Presence of fruit pests within blackberry rows. (**A**) Average number of green June beetles (*Cotinis nitida*) and Japanese beetles (*Popillia japonica*) per row measured by visual survey on two occasions. Average number of bird fecal droppings on blackberry plants per row measured immediately after removal of exclusionary netting; (**B**) Average number of SWD per row captured in baited traps on week 4 alone; (**C**) Average number of adult SWD that emerged from 20 blackberries grown under fine-mesh exclusion and organic spinosad spray management when reared out in laboratory.

**Figure 4 insects-10-00249-f004:**
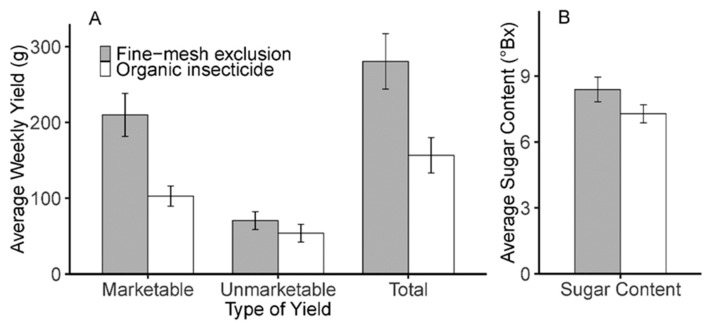
Marketable, unmarketable, total yield, and sugar content of blackberries. (**A**) Average marketable, unmarketable, and total yield of blackberries harvested each week grown under fine-mesh exclusion and organic spinosad spray management; (**B**) Average sugar content of blackberries grown under fine-mesh exclusion and organic spinosad spray management.

**Table 1 insects-10-00249-t001:** Analysis of baited trap captures for *Drosophila suzukii* and all other Drosophilids in blackberry rows under fine-mesh exclusion versus organic insecticide management. Both analyses were performed as linear mixed models using untransformed SWD and square root transformed Drosophilid data. Exclusion treatment describes the effect of fine-mesh exclusion versus insecticide treatments on the number of flies captured. Week describes the effect of trapping week on the number of flies captured. Exclusion Treatment x Week describes the significant interaction uncovered between these two fixed effects.

**1A.**	***D. Suzukii* in Traps**		
	**Estimate ± SEM**	**t**	***p*-Value**
Exclusion Treatment	2.2 ± 1.6	1.3	0.2048
Week	1.8 ± 0.4	4.3	0.0004 ***
Exclusion Treatment × Week	−1.5 ± 0.6	−2.6	0.0201 *
**1B.**	**Drosophilids in Traps**		
	**Estimate ± SEM**	**t**	***p*-Value**
Exclusion Treatment	−9.1 ± 1.5	−6.0	<0.0001 ***
Week	−2.4 ± 0.4	−6.3	<0.0001 ***
Exclusion Treatment × Week	−1.9 ± 0.5	3.5	0.0031 **

Indication of significant effect *** *p* < 0.001, ** *p* < 0.01, * *p* < 0.05.

**Table 2 insects-10-00249-t002:** Analysis of the effect of fine-mesh exclusion versus organic insecticide management on *Drosophila suzukii* infestation of fruits, beetle, and bird presence within blackberry rows, and blackberry fruit yield. Analyses of *D. suzukii* emergence from harvested blackberries, green June beetle presence, and Japanese beetle presence were performed as a linear mixed model on log transformed data. Analyses of bird presence was performed as a paired t-test. Analysis of blackberry sugar content was performed as a linear mixed model on untransformed data. Analyses of blackberry yield was performed as a linear mixed model on square root transformed data. * Bird intrusion analysis was achieved with a paired t-test and therefore does not have a parameter estimate.

	**Berry Incubation Emergence**		
	**Estimate ± SEM**	**t**	***p*-Value**
*D. suzukii* emerged	−3.6 ± 0.5	−7.3	<0.0001 ***
	**Visual Surveys**		
	**Estimate ± SEM**	**t**	***p*-Value**
Green June beetle	−2.2 ± 0.6	−3.9	0.0182 *
Japanese beetle	−2.7 ± 0.7	−3.9	0.0606 ^•^
Bird intrusion *		5.1	0.036 *
	**Berry Production**		
	**Estimate ± SEM**	**t**	***p*-Value**
Sugar content	1.1 ± 0.6	1.9	0.0655 ^•^
Total yield	4.0 ± 0.9	4.5	<0.0001 ***
Marketable yield	4.1 ± 0.8	5	< 0.0001 ***
Unmarketable yield	0.9 ± 0.6	1.4	0.1544

Indication of significant effect *** *p* < 0.001, ** *p* < 0.01, * *p* < 0.05, • *p* < 0.1.

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
