# Peer review of "Effects of Fine-Mesh Exclusion Netting on Pests of Blackberry"

_insects, 2019, doi:10.3390/insects10080249_

Round 1

Reviewer 1 Report

The subject of the article is of extreme relevance, but unfortunately, the article presents some lack of greater scientific soundness.

The main flaws are

1) Better detailing in the sampling area and its surroundings,

2) A very small sample of fruits

3) The collection is only one harvest year

4) Lack of further literature review on SWD

5) Lack of a more detailed and robust statistical analysis

Author Response

On behalf of the authors of this publication, thank you very much for your time reviewing our manuscript. We appreciate your helpful critiques and value the points you have made. We have made changes to this manuscript based on your feedback and have detailed these changes below.

Point 1. Better detailing in the sampling area and its surroundings 

Response 1. We certainly agree that we must supply more information to readers regarding the location of our experiment, its surrounding landscape, and surrounding cropping systems. We have included lines that describe this setting beginning on line 111.

Point 2. A very small sample of fruits

Response 2. We agree that the sample size of our study is small. When designing this experiment, we understood the size limitation of our chosen plots and did consider increasing the sample size by included more, smaller plots per row.  However, we opted for the larger 20m row treatments because larger, whole row plots better mimic how a farmer would realistically apply each treatment. These long rows of caneberries provide easier access to each plant under the netting for harvesters. While more smaller plots would have increased our statistical power, it would have been less realistic for farmers. Furthermore, the fact that we observed clear statistical results with a low sample size suggests that our sample size was adequate to determine statistical differences. We explain our rationale beginning on line 120.

Point 3. The collection is only one harvest year

Response 3. Our study does encompass only one year of harvest. We understand that experimental repetition across multiple years is a powerful way to resolve seasonal differences. We decided to submit this manuscript for publication for two reasons. 1. We observed strong statistical difference in yields of marketable fruits, SWD, and other pests between our two treatments. 2. We believe it is urgent to find management strategies to suppress SWD for berry growers, and especially organic growers, who have few viable management options. We feel this study should contribute to a growing body of evidence suggesting netting exclusions are an effective option.

Our major aim was to test two management regimes that could currently be in use by organic berry producers in plant hardiness zone 6. We believe we have shown that a management regime using the best organic pesticide currently available in our region can be bested by a fine-mesh netting exclusion.  

Point 4. Lack of further literature review on SWD

Response 4. We do hope to accurately describe our manuscript’s place within the current literature. The editor has brought two 2019 publications to our attention [Ebbenga et al, 2019] in grape (Line 350) and [DiGiacomo et al, 2019] (LINE 85) an economic analysis of raspberry losses. We have added a summary of these relevant caneberry publications plus [Leach et al, 2016] (Line 379). 

Point 5. Lack of a more detailed and robust statistical analysis

Response 5. We firmly believe that the suite of statistical analyses we performed are sufficient to show significant patterns within this data. We have adjusted the text within our methods section to streamline the description of the analyses we performed. 

Reviewer 2 Report

Dear authors,

I have appreciated your work. It represents a very simple work, but the worth of a paper does not depend on the complexity of the experimental design.I have reported more specific issues in the pdf version of the manuscript. You should give a more detailed description of your beetles sampling and further, you should consider, in your results discussion, the fact that only one trap in each row, could had produced the lost of more detailed information.  

Author Response

On behalf of the authors of this publication, thank you very much for your time reviewing our manuscript. We appreciate your helpful critiques and agree with each point of your review.  We thank you immensely for your appreciation of this simplistic, but potentially impactful experiment. We have made changes to this manuscript based on your feedback and have detailed these changes below. 

Point 1. You should give a more detailed description of your beetles sampling

Response 1. We have expanded our explanation of our visual sampling protocol starting on line 193 reading as follows: “To document the abundance of Japanese beetles and green June beetles on blackberry canes, we used visual surveys. An observer, starting from one end of each experimental row, carefully moved slowly under the exclusion netting or adjacent control plots choosing the largest fruit-bearing cane per plant on six total plants per row. This was done in a way to minimize disturbing the beetles and avoid double counting beetle individuals. Each fruit-bearing cane was scanned from top to bottom for the number of each beetle species present on fruit, stems, or leaves. This survey was conducted for each row on two occasions on July 17th and again on July 25th. To compare the effects of each management system, we log-transformed beetle abundance and conducted LMM with management treatment and sampling week as fixed effects. Each row was treated as a random effect to account for multiple repeated measurements across weeks. Each proximal pair of rows was treated as a random effect to block row pairs together.”

Point 2. consider, in your results discussion, the fact that only one trap in each row, could had produced the lost of more detailed information

Response 2. Purportedly, SWD tend to frequently move into and out of fruit crop fields. This spastic movement made us confident that one trap would be a sufficient indicator of SWD presence within each field. We have included a section on our choice of experimental design beginning on line 120 within the Methods section. We do, however, agree that the addition of multiple traps within each row would be an excellent addition to future studies within our work. We have added line 416 as a discussion of this future possibility. 

Reviewer 3 Report

Dear Authors

I appreciate your manuscript “Effects of fine-mesh exclusion netting on pests of blackberry” as an alternative to insecticide applications, it is well written but, in my opinion,, it reports a test carried out on a small scale and on a topic already discussed for a long time. Even the aspects reported in the conclusions are topics already read in other articles (about SWD) and already implemented in the normal farmer use. For these reasons, I consider the article unsuitable for the publication in Insects.

Here are some suggestions for improvement

General comments:

-       In different parts of the MeM section is wrote: “We compared XYZ with LMM with management treatment as a fixed effect and row pair, sampling week, and blackberry variety as random effects within the model. We confirmed the normality of the data with a Shapiro-Wilk normality test of model residuals.” I think you can use this sentence at the end and refer it to all the investigated parameter, underlying the differences where present.

-       Why haven't you monitored the temperature and humidity trends inside and outside the nets?

-       Why did you get SWD and other Drosophila species in such small rows? Wasn't the management of the nets correct (as underlined in the conclusion)? or (having closed the nets with some already colored fruit) you had trapped SWD eggs?

Minor points:

-       in line 218, the table is named “S2” but in the text, the reference is to table 2. Is this table the table referred in the text or is in additional materials (as deduced from the code S)?

-       more care in the images (that seems very old as graphics) is needed

Author Response

On behalf of the authors of this publication, thank you very much for your time reviewing our manuscript. We appreciate your helpful critiques and agree with each point of your review.  We thank you immensely for your appreciation of this simplistic, but potentially impactful experiment. We have made changes to this manuscript based on your feedback and have detailed these changes below. 

Point 1.  It reports a test carried out on a small scale  

Response 1.  As mentioned above, we agree that the sample size of our study is small. When designing this experiment, we understood the size limitation of our chosen plots and did consider increasing the sample size by included more smaller plots per row.  However, we opted for the larger 20m row treatments because larger, whole row plots better mimic how a farmer would realistically apply each treatment. These Long rows of caneberries provide easier access to each plant under the netting for harvesters. While more smaller plots would have increased our statistical power, it would have been less realistic for farmers. Furthermore, the fact that we observed clear statistical results with a low sample size suggests that our sample size was adequate to determine statistical differences. We explain our rationale beginning on line 120.

Point 2. On a topic already discussed for a long time. Even the aspects reported in the conclusions are topics already read in other articles (about SWD) and already implemented in the normal farmer use.

Response 2. We agree that there are now a number of studies focusing on exclusionary netting for protection of berry crops, however we believe we will be among the first to publish on applying this treatment to the production of blackberries. We are also among the first to show significant increases in yield for blackberries grown under exclusion netting. We believe these two novel factors merit the publication of our manuscript. Further, at least in the US, still few farmers have adopted protection netting in berry crops. For example, a survey of 40 New England blueberry growers showed that only 30% had ever used bird netting and only 2.5% had ever used insect exclusion netting (see Link 2014, https://www.uvm.edu/vtvegandberry/SWD/LinkThesis3-1-15.pdf). In our region of the US (the southern region), discussions with 17 caneberry growers revealed none used any netting exclusion.

Point 3. I think you can use this (Description of Shapiro-Wilk Normality Test) sentence at the end and refer it to all the investigated parameter, underlying the differences where present.

Response 3. Thank you for suggesting this edit. We have made these changes to the manuscript. 

Point 4. Why haven't you monitored the temperature and humidity trends inside and outside the nets?

Response 4. We agree that the temperature and humidity within our experimental rows would be an appropriate piece of information that we should have collected for this publication. The major effect of both temperature and humidity underneath or outside of exclusion is to speed up the ripening process of fruits.  Because we pursued fruit harvest until all canes ceased producing for the year, we believe that the lack of this data does not significantly change the meaning of our findings.

Point 5.  Why did you get SWD and other Drosophila species in such small rows?

Response 5. Our experiment did undergo a brief period in which gaps were present within our exclusion netting. These gaps were caused by high winds one night during the experimental duration, but were resolved within 12 hours. It is our strong belief that SWD entered into two of our exclusion rows during this window of opportunity and then began to reproduce as we approached the end of our experimental sampling. We have added Line 142 to our manuscript within the Methods that explains this event for readers. However, even given these strong winds, we still demonstrated significant differences in SWD infestation and abundance in traps.

Point 6. The table on Line 218 is named S2 erroneously

Response 6. We appreciate your catching this error for us. We have changed the text within the manuscript. 

Point 7. More care in the images (that seems very old as graphics) is needed

Response 7. We do agree that our graph design could be improved. We have worked in R to modernize our figures to make data more readily legible. 

Round 2

Reviewer 1 Report

The manuscript is good now. 

Author Response

Thank you very much for your work reviewing our manuscript! We appreciate your comments and know that our manuscript is stronger now after making edits based on your reviews. 

In addition to your comments, we have edited the grammar and spelling of manuscript slightly throughout its length to allow for greater readability.

We have also edited the x-axis labels on figure 3 to use the Latin names of the three pest species. This uses fewer characters and better spaces-out the labels to allow for greater readability as well. 

Once again, thank you so much for your time!

Sincerely, 

Ryan Kuesel

Reviewer 3 Report

Dear Authors,

I read with pleasure your answers to my questions/points and the changes you made throughout the text. Now, I think the work is well improved and more detailed. The figures also look better.

I perfectly agree with this sentence: “Additional in-depth economic and agronomic analyses will be required to determine whether it is more profitable to switch from current organic pest control practices to exclusionary netting in blackberry production. Future studies should look to compare the costs of both the material inputs and labor required by both management schemes.”

Regarding the problem with severe weather event, you could also advise farmers to install some monitoring traps inside the nets to understand early if any SWD has entered the nets. This could also help in reducing the SWD population inside the nets. Or you can suggest an application of a knock-down insecticide such as the Spinosad you mentioned.

For this reason, I hope your manuscript will soon be published in Insects.

Best Regards

Author Response

Thank you very much for your work reviewing our manuscript! We have appreciated your comments and know that our manuscript is stronger now after making edits based on your reviews. 

We agree that the inclusion of a statement to producers that baited traps within exclusionary treatments is important, so we have added lines beginning on line 278 to do so. 

Once, again we thank you so much for your time!

Sincerely, 

Ryan Kuesel